# Dietary and Nutritional Interventions for the Management of Endometriosis

**DOI:** 10.3390/nu16233988

**Published:** 2024-11-21

**Authors:** Nour Abulughod, Stefanie Valakas, Fatima El-Assaad

**Affiliations:** 1University of New South Wales Microbiome Research Centre, School of Clinical Medicine, UNSW Medicine & Health, St George & Sutherland Clinical Campuses, Sydney, NSW 2217, Australia; 2The Dietologist, Sydney, NSW 2008, Australia

**Keywords:** endometriosis, diet, nutrition, dietary intervention, nutritional intervention, microbiome, microbiota, inflammation, estrogen, antioxidants

## Abstract

Endometriosis is a chronic, complex, systemic inflammatory condition that impacts approximately 190 million girls and women worldwide, significantly impacting their quality of life. The effective management of endometriosis requires a multi-disciplinary and holistic approach, one that includes surgical and medical management, such as a laparoscopy and a chronic medical management plan, as well as dietary, nutritional, and lifestyle adjunct interventions, such as pelvic pain physiotherapy and acupuncture. There is growing evidence to support the role of dietary and nutritional interventions in the adjunct management of endometriosis-related pain and gastrointestinal symptoms. However, the implementation of these interventions is often not regulated, as patients with endometriosis often adopt self-management strategies. Diet and nutrition can modulate key players integral to the pathophysiology of endometriosis, such as, but not limited to, inflammation, estrogen, and the microbiome. However, it is unclear as to whether diet plays a role in the prevention or the onset of endometriosis. In this review, we discuss three key players in the pathogenesis of endometriosis—inflammation, estrogen, and the microbiome—and we summarize how diet and nutrition can influence their mechanisms, and consequently, the progression and manifestation of endometriosis. There is a major need for evidence-based, non-invasive adjunct management of this debilitating disease, and diet and nutritional interventions may be suitable.

## 1. Introduction

Endometriosis is a complex, systemic, chronic inflammatory gynecological disease [1] that affects approximately 10% of women of reproductive age worldwide [2], 2–4% of perimenopausal women [3,4], and up to 50% of infertile women [5]. Endometriosis is defined as endometrial-like epithelium and/or stroma growing outside the endometrium and myometrium and presents with chronic inflammation [6]. Endometriosis can be classified into three subtypes: (i) superficial: small, flat, shallow lesions present on the pelvic cavity, accounting for approximately 80% of endometriosis; (ii) ovarian: cysts or endometrioma that form within the ovaries; and (iii) deep: infiltrating 5 mm or more beneath the peritoneum [6,7]. Although rare, extra pelvic endometriosis (endometriosis found outside the pelvis) has been found in the intestines, bladder, diaphragm and thoracic wall, abdominal wall, and central and peripheral nervous system [7,8]. Endometriosis is predominately ‘benign’; however, there is increasing evidence to support its malignant transformation and increased predisposition to ovarian cancers [9,10].

Symptoms often vary with patients, and their presentation can also fluctuate over time, making it difficult to diagnose efficiently and hence contributing to the prolonged time to diagnosis [11]. The main symptoms include pelvic pain, painful periods (dysmenorrhea), painful intercourse (dyspareunia), painful defecation (dyschezia), painful urination (dysuria), fatigue, depression, and irritable bowel syndrome (IBS) symptoms, such as constipation and/or diarrhea and abdominal discomfort [7]. The symptoms of endometriosis and other conditions, such as IBS (a functional gastrointestinal condition) and autoimmune conditions, such as Inflammatory Bowel Disease (IBD), can overlap [12,13]. Studies have shown that IBS was two or three times higher in endometriosis patients [14,15], and women with endometriosis had an increased risk of IBD [16]. These correlations have not yet been fully understood, although they do present a significant diagnostic challenge, and awareness of the co-morbidities between endometriosis and other conditions is important.

Pain is the most common overlapping symptom between endometriosis, IBS, and IBD [17]. The visceral hypersensitivity commonly found in these conditions exacerbates the pain. Through afferent and efferent nerve reflex arches, pain fibers in visceral organs may be linked through a reflex arc called viscero-visceral hyperalgesia, transmitting painful stimuli [18,19]. This, coupled with the frenetic activity of low-grade inflammation triggered by mast cell (MC) activation, immune system dysfunction, intestinal permeability, dysbiosis (imbalance of the microbiome) and host–microbiota interactions, neuronal inflammation, hormone dysregulation, and common genetic variables, may account for the interrelationship between IBS, IBD, and endometriosis [15,17,20]. Even in the absence of endometrial bowel involvement, gastrointestinal symptoms still occur in patients with endometriosis, suggesting the condition may indirectly affect the enteric nervous system, causing subsequent changes in visceral sensitivity.

There is an increasing need to move away from the traditional gold standard for diagnosing and managing endometriosis, which relied on direct visualization during surgical laparoscopy and excisional biopsy with histological confirmation [21]. Instead, there is a push to incorporate non-invasive diagnostic tools and adjunct treatments [22,23]. Because the disease is progressive and can vary wildly in its presentation over the course of a patient’s life (e.g., adolescence, adulthood, perimenopause), it can make diagnosis and treatment challenging. There are currently no known clinical diagnostic biomarkers for endometriosis, although some novel microbial, metabolite, and microRNA biomarkers are beginning to emerge [24,25].

The current medical treatment aims to treat endometriosis-related pain, and it includes progestins, nonsteroidal anti-inflammatory medicines (NSAIDs), the combined oral contraceptive pill (COC), and GnRH agonists [21,26]. Other complementary hormonal therapies are also available, such as levonorgestrel-releasing intrauterine device, Dienogest (a progestin medication), and aromatase inhibitors (block estrogen production) [21]. However, both the surgery and complementary therapies do not always prevent the recurring growth of endometrial lesions or reduce pain and other unwanted gastrointestinal symptoms, such as bloating and digestive issues. The surgical laparoscopy is essential to remove lesions [27], and the use of robotic-assisted laparoscopic surgery is becoming more popular [28]. However, endometriosis can often be asymptomatic, and surgery is more complete in early-stage endometriosis, which cannot be detected on ultrasound [22]. In addition, surgical removal of the lesions does not guarantee that the pain and lesions will not return. The overall recurrence rates of endometriosis after surgery range between 6 and 67% [29]. Most women have used at least three different medical treatments to manage their symptoms, and almost 20% continued to use them for ten years or more, with a discontinuation rate between 35 and 50% due to ineffectiveness or side effects [30]. While these treatment regimens are crucial in managing the pain associated with endometriosis, they do not necessarily work for every patient, and their effectiveness can vary. Endometriosis can have a significant detrimental effect on the physical, mental, sexual, and social well-being of the affected patients [31,32]. Patients with endometriosis may experience stress from recurrent surgical procedures and medical therapies, as well as from their side effects and the psychological impact of repeated treatment failures.

There is an increased usage of self-management approaches in women with endometriosis [33]. Changes to diet had the highest self-reported ratings for the ability to reduce pelvic pain and gastrointestinal symptoms, although no single diet was reported to manage symptoms better than others [33,34,35,36,37]. A key gap in knowledge is whether diet and nutrition can play a role in the mitigation of symptoms and potentially the progression of endometriosis. A recent study showed that adherence to a healthier dietary pattern, higher in fruits and vegetables and lower in red meat and trans fats (Alternative Healthy Eating Index), was associated with a 13% lower risk of endometriosis diagnosis and most likely impacted pelvic pain [38]. However, there is no consensus on evidence-based clinical guidelines for diet and nutrition in endometriosis, and this is an area in research that is underdeveloped [39]. Despite increasing research outputs in the field of diet and nutrition, there is still a need for well-designed randomized controlled studies as well as more original research on the therapeutic potential and safety of specific diets and nutrients in the management of endometriosis [36,39].

### Aim

This study reviewed the existing literature on dietary and nutritional interventions for endometriosis. We examined key factors involved in its pathogenesis, including inflammation, estrogen, and the microbiome, and explored how diet and nutrition might influence these processes. We also discussed current research on dietary interventions for managing endometriosis, identified gaps in the research, and suggested directions for future studies.

## 2. Methods

A search of the literature was conducted through PubMed and Embase databases. An advanced search included papers from the past ten years, i.e., from 2013 to 2024, and only papers in English were included. Research in both animals and humans was included. Studies mentioned in the references of these articles were also retrieved, reviewed for relevance, and only then included in the reference list, even if they were older than ten years. Abstracts alone were not included, and only those published as full-length articles were included in the reference list. The search query included keywords either alone or in combination with endometriosis, such as [Endometriosis] “and” [diet]. The terms were searched in PubMed (meSH). Keywords included red meat, dairy, resveratrol, quercetin, FODMAP diet, Mediterranean, high fiber, anti-inflammatory, mast cells, omega 3, lifestyle, microbiome, nutrition, antioxidants, anti-inflammatory, gluten-free, estrogen, carbohydrates, omega 6, vitamin D, milk, immune, inflammation, pro-inflammatory, nickel-diet, probiotics, ginger, n-acetylcysteine, curcumin, turmeric, vitamin C, anti-histamine, gut, vitamin E, medical treatment, dietary inflammatory index, irritable bowel syndrome, pathophysiology, etiology, dysbiosis, processed food, and saturated fats.

## 3. Inflammation, Estrogen, and the Microbiome in the Pathogenesis of Endometriosis

The key players underlying the pathogenesis of endometriosis include, but are not limited to, inflammation, estrogen, and the microbiome, with the microbiome emerging as a significant new area of research in the progression of this disease [40,41,42]. These three key players are modifiable by diet and nutrients, but to what extent and how is yet to be explored comprehensively and definitively. These players also offer the possibility of targeted interventions through diet and nutrition to modulate their role and, consequently, modulate the impact of endometriosis, particularly on endometriosis-related pelvic pain and infertility.

### 3.1. Inflammation

The involvement of the immune system in the initiation and progression of endometriosis is widely accepted [43]. Immune cells, especially T and B lymphocytes and natural killer (NK) cells, seem to play a significant role in the progression of endometriosis. These cells, as well as macrophages, monocytes, and mesothelial cells, have been found in higher concentrations of peritoneal fluid sampled from patients with endometriosis [44]. In addition, the activation of these cells is more pronounced in patients with endometriosis [44].

In addition to the involvement of the immune system, oxidative stress also contributes to the pathophysiology of endometriosis [43] and endometriosis-related infertility [45]. Higher levels of iron, ferritin, and hemoglobin have been detected in the peritoneal fluid of women with endometriosis [46]. Iron overload in endometriosis could lead to the development of oxidative stress [47]. High oxidative stress could cause an inflammatory response leading to the destruction of the peritoneal mesothelium and eventually forming adhesions [44,46,47]. Oxidative stress can amplify epigenetic mechanisms, such as DNA methylation alterations, which have been implicated in the pathogenesis of endometriosis [48]. There is emerging evidence to support endometriosis as an epigenetically regulated disorder [49]. Oxidative stress caused by hyper-homocysteinemia can be triggered by diets high in methionine (precursor of homocysteine)-rich foods, like red meat and dairy, or disorders of folate metabolism caused by methylenetetrahydrofolate reductase (MTHFR) gene polymorphisms. MTHFR C677T homozygous polymorphisms might be considered a risk factor for endometriosis [50,51]. In addition, nutrients such as folic acid, vitamin B12, choline, and B6 present in foods participate in the one-carbon unit cycle of the body, and one-carbon units can be methyl donors and may impact the DNA methylation process [52]. It is these meandering relationships that impact gene expression coupled with the persistent release of inflammatory agents, such as cytokines, reactive oxygen species (ROS), and autoantibodies [53], that can cause chronic inflammation. Over time, chronic inflammation could be detrimental [54] and can lead to MC activation, dysregulation of nociceptive endings, dysbiosis, altered intestinal permeability, and translocations of bacterial endotoxins [53,55].

MCs release several different molecules, including prostaglandins, histamine, interleukins, leukotrienes, and nerve growth factor (NGF). The neurotrophic role of the NGF released from MCs enhances neuronal fiber activation. Moreover, the pro-inflammatory actions of these molecules activate nociceptors, causing vasodilation and edema, resulting in visceral pain [53,56]. Activation of MCs found in both conditions appears to be located near the nerve endings in both the pelvic and abdominal regions, as well as the bowel mucosa and its immunological connection [15,53]. Furthermore, macrophages are activated to release more cytokines, prostaglandins, and complement components, initiating an inflammatory response [44]. A high number of activated MCs have been found near endometriotic lesions close to nerve fibers [57]. MCs have been shown to switch on afferent nerve fibers of the nociceptors, activating inflammatory markers and causing a release of neurotransmitters, such as histamine, prostaglandins, and substance P (SP) [56]. This may contribute directly to the pain associated with endometriosis.

### 3.2. Estrogen

Endometriosis is a hormone-dependent condition; estrogen and estrogen receptors [58] play a major role in its pathogenesis [7]. There are several estrogen receptors (ERs), such as estrogen receptor alpha (ERα) and estrogen receptor beta (ERβ), both functioning as nuclear transcription factors. The main role of ERα is to regulate cell growth-related genes, whereas ERβ plays a significant part in the progression and apoptosis of the cell cycle [58,59]. The key hormone involved in the development and persistence of endometriotic tissue, as well as the pain and inflammation correlated with it, is 17β-Estradiol (E2). E2 is mostly produced locally in the endometriotic tissue; however, E2 can also reach endometriosis through circulation. The accumulation of E2 is thought to play a crucial role in the progression of endometriotic lesions by binding to and activating the ERs [60].

In the unaffected endometrium, ERα, a key mediator of estrogenic activity, is considerably higher than Erβ [60]; however, women with endometriosis exhibit higher levels of ERβ expression in their eutopic endometrium, which is associated with increased inflammation, enhanced cell proliferation, suppressed apoptosis, and pain transmission [61]. Additionally, deeper dyspareunia and moderate to severe dysmenorrhoea were predicted by a higher expression of Erα; this suggests that in addition to circulating estrogens, tissue expression of ERα is also associated with the severity of the symptoms [62].

Furthermore, early initiation of endometriosis could be a result of crosstalk that occurs between ERα and interleukin (IL) 6. As a result, activated estrogen receptors could induce an inflammatory response. Estrogen can bind to MC ERα and consequently may promote MC activation and subsequently inflammation [56]. Therefore, high levels of estrogen endometriotic lesions provide the perfect setting for the recruitment of MCs [63,64].

Estrogens consist of a variety of hydroxylated (OH) and methoxylated (MeO) catechol estrogen (CE) metabolites with multiple biological actions. The synthesis of CE metabolites from estradiol (E2) and estrone (E1) includes different metabolic processes, notably the 2-hydroxylation (2OH), 4-hydroxylation (4OH), and 16-hydroxylation (16OH) pathways, and the activity of catechol-O-methyltransferase (COMT) to generate 2- and 4-MeOCEs [65]. The (2OH) metabolic pathway is elevated in patients with ovarian endometriosis and is associated with an increased likelihood of pain [65]. Therefore, an environment higher in estrogen may be associated with pain severity [66].

Disruption in estrogen and progesterone signaling in endometrial lesions can lead to progesterone resistance and unopposed estrogen [67]. This imbalance causes an upregulation of estrogen, which could induce cell proliferation as well as inflammation. This process may, therefore, stimulate the progression of lesions and impair endometrial receptivity, potentially worsening the pain symptoms associated with endometriosis [67].

### 3.3. The Microbiome

The dynamic and diverse collective genomes of bacteria, archaea, viruses, and fungi that reside in and on our body make up the microbiome [68,69,70]. The human microbiome plays a role in varied physiological functions, including immune system development, defense against pathogens, host nutrition, and the production of short-chain fatty acids (SCFAs), which are essential for various metabolic processes, including the host metabolism of energy, vitamin synthesis, and fat storage [69,71]. A disruption in the microbiome, dysbiosis, has been linked to several pathophysiological conditions [72,73]. Importantly, the microbiome can be modified by diet, with varied diets influencing the composition of the microbiome [74,75,76], and gut-microbiota modulating nutritional-derived therapies have potential to promote health [77].

Our previous work has shown that the microbiome is emerging as a key player in endometriosis [24,40,41]. Several additional studies have demonstrated a relationship between microbiota and endometriosis in patients [78,79,80,81]. Patients with endometriosis have more pathogenic species in their oral, vaginal, stool, and cervical microbiome [24,72]; for example, *Escherichia*, *Enterococcus*, and *Tepidimonas* were found to be notably higher in the vaginal microbiota [24]. However, in a cohort study that included 136 women with endometriosis, no notable changes in the microbiome were observed between women with or without endometriosis [82].

*Fusobacterium*, an opportunistic pathogen, was found in the oral samples of those with mild to severe endometriosis [24] and has been found in the endometrium and endometrial lesions of endometriosis patients [83]. Infection with *Fusobacterium* is being debated as a potential cause of endometriosis [84,85]. However, focusing on a single bacterium causing endometriosis is increasingly being challenged by our growing understanding of the collective activity of microbial communities and their complex interactions with the host.

The gut microbiome is capable of metabolizing circulating estrogens and is referred to as the estrobolome [86]. This is regulated through secretions of β-glucuronidases, enzymes that deconjugate estrogens into their active forms. When dysbiosis occurs, deconjugation decreases, resulting in reduced circulating estrogens. Furthermore, conjugated estrogen and phytoestrogens secreted through the bile acid are deconjugated by the bacterial production of β-glucuronidase. These “active” estrogens that are unbound are reabsorbed by the gut and enter the bloodstream, where they act on estrogen receptors. An imbalance in the microbiota disrupts this homeostasis, leading to an increased level of estrogen metabolites. Endometriosis patients may have an abundance of β-glucuronidase-producing bacteria in their gut microbiome, which may, therefore, increase the levels of estrogen metabolites, stimulating epithelial proliferation [87]—a state commonly found in the reproductive tract of women with endometriosis [86].

## 4. Dietary and Nutritional Interventions for the Management of Endometriosis

There is growing evidence to suggest that diets and nutrients may modulate the pathophysiological processes underlying endometriosis, such as inflammation, estrogen pathways, and the microbiome and microbial metabolite interactions (Figure 1) [39,88]. Interest in dietary and nutritional interventions for the management of endometriosis is driven by patient-seeking strategies [33,37]. Evidence for dietary and nutritional interventions is limited due to the low number of clinical trials and patient-reported outcome studies investigating dietary interventions for the management of endometriosis [39].

Nutritional deficiencies or dietary exposures have been associated with an increased risk of endometriosis. For example, women who presented with a pro-inflammatory diet, such as those with a high intake of sugar, highly processed foods high in trans fatty acids, and saturated fat intake, were four times more likely to have endometriosis, and women consuming more than two servings per day of red meat showed a 56% higher risk of endometriosis [90,97]. Interestingly, an earlier study of Italian patients with laparoscopically confirmed endometriosis found that consumption of milk, liver, carrots, cheese, fish, and whole-grain foods, as well as coffee and alcohol consumption, were not significantly related to endometriosis [98]. In addition, Vitamins C and E were lower in the follicular fluid of women with endometriosis [99].

Research shows that diet may play an important role in the treatment, management, and even prevention of endometriosis, with some studies showing it can positively influence pain perception in women with endometriosis [100]. We discuss the main diets studied in the pathogenesis and management of endometriosis below. We also provide a summary of some available diets (Table 1) and nutrients (Table 2) evaluated in the endometriosis research.

### 4.1. Diets

#### 4.1.1. FODMAP Diet

FODMAP (fermentable oligosaccharides, disaccharides, monosaccharides, and polyols) diet is a dietary approach whereby these short-chain carbohydrates are restricted in people who may poorly absorb them and are easily fermented by bacteria [125]. In individuals with visceral hypersensitivity, their osmotic activities and gas production may result in intestinal luminal distension, causing discomfort and bloating that may lead to pain. Often, women who are diagnosed with endometriosis experience gut symptoms like those of IBS. Women with endometriosis are two to three times more likely to be diagnosed with IBS [14,17]. A high FODMAP diet has been shown to cause colonic barrier loss and MC activation, as shown in Figure 1. The low FODMAP diet showed a reversal in those pathophysiological changes [126]. Moreover, a low FODMAP diet showed significant improvement in gut symptoms in patients with endometriosis [18,37,127], including a 50–80% symptomatic relief in patients with IBS [128,129,130]. In one retrospective study, out of 160 women with IBS who met the Rome III criteria, 36% had concurrent endometriosis. Of those, 72% of the women with endometriosis reported that following a low-FODMAPS diet helped to relieve their gastrointestinal symptoms [18]. Recommending a low FODMAP diet to improve gut-related symptoms may be an option for patients with endometriosis. The FODMAP diet is best adopted under the supervision of a dietician and other healthcare practitioners, such as a dietician or a naturopath [131]. A dietitian is best able to identify triggers and support the reintroduction of specific food groups in a methodical way [132].

#### 4.1.2. Gluten-Free Diet

Gluten, the elastic network between the proteins gliadin and glutenin, is found in many types of grains and can trigger inflammatory responses in some individuals. A gluten-free diet has shown some potential to improve pelvic pain and gastrointestinal symptoms [100,102,133]. Gliadin proteins may increase MCs, stimulating an inflammatory response; this may be why some women experience an improvement in abdominal pain when eliminating gluten [102,103,134]. A retrospective analysis from a cohort of 363 laparoscopically diagnosed endometriosis patients found that the removal of wheat or gluten showed an improvement in pain scores, and when wheat was reintroduced, symptoms such as pain, bloating, diarrhea/constipation, headaches, and fatigue recured [135]. In a 2021 observational study involving 157 participants, the addition of vegetables, an increase in fiber, and a reduction of gluten, meat, caffeine, saturated fats, and chocolates by 50% in addition to a 30% reduction in dairy products showed a decrease in pain associated with endometriosis [34]. The limitations of this study include the finding that no single food item or nutrient was correlated to an improvement in symptoms; rather, a combination of foods was, and no specific dietary modification was linked to a higher quality of life. However, according to one study, gluten is not likely to be a contributing cause or symptom of endometriosis [136]. Research on the association between gluten and endometriosis has been limited, and further studies need to be conducted in this field. Despite the evidence to suggest that specific patient groups benefit from a gluten-free diet, research suggests that there are possible nutritional deficiencies and adverse psychological effects associated with a gluten-free diet. Additionally, the cost of gluten-free products is higher than that of their gluten-containing equivalents [137].

#### 4.1.3. Mediterranean Diet

The Mediterranean diet comprises high consumption of cold-pressed extra-virgin olive oil and vegetables, including leafy green vegetables, fruits, cereals, nuts, and pulses/legumes; moderate consumption of fish, poultry, and dairy products; and low consumption of eggs, red meat, and sweets [138,139]. In a single-arm trial conducted in Austria, the impact of the Mediterranean diet on the pain related to endometriosis was investigated [36]. Sixty-eight women who had previously undergone a laparoscopic diagnosis of endometriosis were included. Participants were required to follow a strict diet regimen that included cold-pressed oils, fresh vegetables and fruit, white meat, fish high in fat, soy products, whole-meal products, and foods high in magnesium. Red meat, sweets, animal fats, and sugary beverages were prohibited during the intervention. Due to pregnancy or switching to conventional treatments, 25 research participants did not follow the suggested diet plan. Nonetheless, all sixty-eight patients were included in the intention-to-treat analysis. Overall, there was a significant improvement in pain, dyspareunia, dyschezia, and dysmenorrhea [36]. On the contrary, in a 2020 case–control study conducted by Ashrafi et al. [140], where a total of 413 women were divided into two groups based on the results of their laparoscopy: a normal pelvis (control group) or endometriosis (case group), a decreased risk of endometriosis was found to be significantly correlated with the consumption of red meat and green vegetables. However, eating more fresh fruit, dairy, cheese, and legumes was linked to a lower risk of endometriosis [140]. It is interesting to note that eating more legumes may be protective. The reliance on self-reported dietary intake introduces a significant risk of recall bias. Participants may not accurately remember or report their past dietary habits, particularly if the reporting is done after the diagnosis of endometriosis, which could lead to misinterpretation of the study. Extra-virgin olive oil, fruits and vegetables, grains, and herbs are major components of the Mediterranean diet and include a variety of compounds, such as antioxidants, polyphenols, and anti-inflammatory compounds, all important factors in endometriosis. It is, therefore, a diet worth considering when managing endometriosis, but definitive studies are still lacking [141].

#### 4.1.4. High-Fiber Diet

A high-fiber diet refers to one that comprises both soluble and insoluble fibers in abundance. This can include fruits, vegetables, whole grains, lentils, legumes, nuts, and seeds [142,143]. A cohort study based on the data obtained from 70,835 pre-menopausal women conducted by Harris et al. found that women who consumed more than one serving of citrus fruit a day had a 22% reduced risk of endometriosis [144]. No correlation was made for the overall vegetable intake; however, the consumption of more than one serving of cruciferous vegetables had a 13% higher risk of endometriosis [144]. Limitations to this study may include self-reporting of dietary intake, which could influence the results [144]. Conversely, a large study conducted by Parazzini et al. showed that the consumption of fruits and vegetables reduced the risk of endometriosis; however, this was another case–control study, which may mean the collected information may not be as accurate [98].

The development and persistence of endometriotic tissue, as well as the inflammation and pain it causes, are all largely regulated by the hormones E2 and ERβ [60,145,146]. Whole-plant diets, a high fiber intake, and a diverse microbiome all help to increase the binding and excretion of sex hormones. A high-fiber diet has been shown to significantly reduce serum estrogen concentrations [105,106,107]. Additionally, a high-fiber diet can downregulate MC activation [147]. Therefore, implementing a whole-food, high-fiber, and plant-based diet, including high fruit and vegetable intakes in addition to fermented foods, has been shown to increase microbiome diversity, influencing the estrobolome in addition to reducing oxidative stress and inflammation [86,107,108,109].

#### 4.1.5. Anti-Inflammatory Diet

In comparison to the Mediterranean diet, the anti-inflammatory diet comprises a larger diversity of fruits and vegetable consumption, including a variety of colors to increase the phytonutrient content, a focus on more fatty fish and some lean animal protein, reduced carbohydrates, whole grains in small amounts, high fiber, olive oil, and some anti-inflammatory spices and herbs, such as turmeric and ginger [148,149]. A recent 2023 large cohort study of 3410 participants found a significant correlation between pro-inflammatory foods and endometriosis risk, suggesting an anti-inflammatory diet is essential. The Dietary Inflammatory Index (DII) score was used in this study. DII scores were assessed for the role of food on inflammatory biomarkers, such as IL-6, C-reactive protein, and TNF-a, all of which increased in endometriosis [150,151]. Another large-scale Japanese study involving 3249 participants assessed the association between an anti-inflammatory diet and preterm birth in women with endometriosis and found a significant reduction in preterm births, low birth weight, and an overall improvement in well-being. While the study focused on preconception care and its improved rates of preterm births, it is important to note that endometriosis is a chronic inflammatory condition, and therefore, women with endometriosis may have improved health outcomes from an anti-inflammatory diet as suggested by this study [110]. Endometriosis presents a wide range of symptoms and severities, and while an anti-inflammatory diet may benefit some patients, it may not be as effective for others. Therefore, the study would benefit from further exploration of how such a diet specifically interacts with various stages of endometriosis.

#### 4.1.6. Low-Nickel Diet

Nickel is a widely distributed element found in soil, water, plants, and animals. Its biological purpose in humans is unclear. However, high amounts of nickel can be toxic [152]. Food groups including dairy, cereals, vegetables, legumes, nuts, and seeds contribute significantly to dietary nickel exposure [152,153]. It is unclear whether there is a correlation between nickel exposure and endometriosis. Following an assessment of Korean women aged 20 to 40 who attended a medical facility between 2009 and 2011, researchers found that 535,818 women did not have endometriosis, while 7259 women did [154]. After adjusting for age and the year of data collection, it was observed that women with endometriosis exhibited a higher rate of nickel allergy [154]. In a smaller study involving 50 women, those with endometriosis were found to have significantly higher whole blood nickel levels. Furthermore, patients who are sensitive to nickel typically experience gastrointestinal symptoms, but these can also be systemic, affecting the skin, neurological system, and reproductive system. In an open pilot study, after three months of following a low-nickel diet, women with endometriosis experienced a statistically significant decrease in all gastrointestinal, extra-intestinal, and gynecological symptoms, including those common to endometriosis (chronic pelvic pain, dysmenorrhea, and dyspareunia) [101].

### 4.2. Dairy

Cow’s milk contains detectable amounts of steroid hormones as well as estrogens, human serum estrone (E1), and progesterone [92,155]. Exposure to estrogens increases the risk of developing breast and endometrial cancers, as they mediate cellular growth and differentiation in both endometrial tissue and mammary glands [156,157,158]. It has been demonstrated that dairy protein influences the rise of insulin-like growth factor-1 (IGF-1) [93,159]. Furthermore, it has been found that excess IGF-1 causes pro-inflammatory cytokines in circulating immune cells [160,161] (Figure 1). While the above study was conducted to assess the risk of endometrial cancer, it is important to note that excess IGF-1 could promote inflammation and pain in endometriosis [94,162]. Contrary to this, there have been several studies demonstrating a positive effect of dairy products with a reduced risk of endometriosis [163,164], and other literature evaluating the links between dairy and inflammation found no correlation [164,165]. In a large nurse’s health prospective study that was followed over 14 years, intakes of calcium, magnesium, and vitamin D from diets (including fortified foods) were found to be negatively correlated with endometriosis. Additionally, a 5% decrease in the risk of endometriosis was linked to an increase in total dairy food intake of one serving per day [166].

### 4.3. Red Meat

In a meta-analysis conducted by Parazzini et al., consumption of red meat, including beef, lamb, mutton, pork, veal, venison, and goat, has been associated with an increased risk of endometriosis in comparison to women with the lowest intake [98]. Interestingly, in a prospective cohort study, women who ate more than two servings of red meat per day had a 56% greater likelihood of developing endometriosis than those who ate less than one serving per week [90]. Possibly due to the growth-promoting sex hormones, both red meats as well as dairy foods may affect the levels of sex hormone-binding globulin (SHBG) and estrogen in the blood [167,168,169]. Furthermore, processed as well as unprocessed red meat has been found to increase inflammation [170,171,172] (Figure 1).

### 4.4. Nutrients

#### 4.4.1. Antioxidants

The body produces antioxidants as a defense strategy to counteract reactive oxygen species (ROS). Oxidative stress develops due to an imbalance between ROS and antioxidants [43,173]. Endometriosis patients have reduced total antioxidant levels, which make them more susceptible to oxidative stress [43]. Through the induction of cytokines and growth factors, oxidation encourages the growth of endometrial-like tissue, thus upregulating inflammation in the peritoneal cavity [44,174]. Furthermore, ROS are increased in endometriotic cells and cellular proliferation. Oxidation could occur due to an accumulation of iron in the peritoneal fluid caused by the presence of endometrial-like cells [175,176]. This process may play a major role in the development of endometriosis [173,177].

#### 4.4.2. Vitamin D

Vitamin D is a fat-soluble prohormone that plays an important role in bone-mineral metabolism, including calcium and phosphorus metabolism and skeletal homeostasis [178]. Cholecalciferol, often known as vitamin D3, is synthesized by sunlight on the skin from 7-dehydrocholesterol, a precursor to cholesterol. It can also be obtained from the diet through animal (cholecalciferol) and vegetable (ergocalciferol) sources [179]. In endometriosis, vitamin D may decrease inflammation and immunoregulation and inhibit angiogenesis [113]. A randomized, double-blind, placebo-controlled clinical trial involving 50 women assessed the effects of a 12-week supplementation with 50,000 IU of vitamin D every two weeks on clinical symptoms and metabolic profiles in women with endometriosis. The findings demonstrated that vitamin D intake led to a significant improvement in pelvic pain. The limitations of this study include a small sample size and the potential impact of surgical treatment of endometriosis before the intervention on the research results [111].

Additionally, in another double-blind, randomized, placebo-controlled study involving 69 participants, vitamin D supplementation in adolescents with surgically confirmed endometriosis resulted in statistically significant improvements in pelvic pain. However, this improvement was not clinically or statistically significant when compared to the placebo group in adolescent girls and young women with endometriosis. A major strength of this study was its rigorous design. Despite participants not being low in vitamin D at baseline, suggesting a potentially stronger effect in a vitamin-D-deficient population, one limitation was the possibly lower concentration of the vitamin D pill compared to the previous RCT [112].

#### 4.4.3. Curcumin

Curcumin, a key polyphenol found in turmeric, a rhizomatous herbaceous perennial from the ginger family, has been extensively studied for its potent anti-inflammatory and antioxidant mechanisms [180,181]. Curcumin suppresses endometrial cell proliferation by reducing the levels of E2 [114]. In addition, treating endometriotic stromal cells with curcumin significantly inhibited the TNF-α-induced secretion of IL-6 and IL-8 and suppression of TNF-α [115].

#### 4.4.4. *N*-Acetyl-Cysteine (NAC)

*N*-acetylcysteine (NAC), also referred to as *N*-acetylcysteine, serves as a precursor to the amino acid L-cysteine and, subsequently, to the antioxidant glutathione (GSH) [182]. It is predominantly found in Allium species, particularly in onions. NAC is listed as one of the 40 Essential Medicines by the World Health Organization and has been a recognized medication since the 1960s. In addition, NAC has been used as an additional therapy for pulmonary, gastrointestinal, and neuropsychiatric conditions [183]. In an observational cohort study on ovarian endometriosis, a total of 92 patients were included, with 47 in the NAC-treated group and 45 in the untreated group [116]. After 3 months, the NAC-treated group showed a slight reduction in cyst mean diameter (−1.5 mm), compared to a significant increase (+6.6 mm) in the untreated group. Additionally, there was a decrease in dysmenorrhea, dyspareunia, and chronic pelvic pain observed in the NAC-treated patients. A limitation of the study is the absence of a placebo group, which prevents the validation of the observed changes in pain incidence and severity [116].

#### 4.4.5. Alpha Lipoid Acid (ALA)

Alpha-lipoic acid (ALA) is an organosulfur molecule found in nature that is produced by humans, animals, and plants. ALA is a substance often found in mitochondria and is required for various enzymatic processes. It is mostly acquired from the diet, particularly meat, vegetables, and fruits. ALA provides multiple health benefits, including its antioxidant properties [184,185]. In an experimental rat model with endometrial implants, ALA was administered for 14 days to evaluate the biochemical and histopathologic parameters. The results showed that the ALA group had significantly lower serum total oxidant status and oxidative stress index levels, reduced endometrial implant volumes, decreased TNF-α levels in serum and peritoneal fluid, and improved histopathologic scores compared to the control group [117]. The limitations of this study include its use of rat models, which may not fully capture the complexities of human endometriosis. Human studies would provide more relevant results, ensuring greater applicability of the findings to clinical practice. In a separate study involving human endometriotic epithelial and stromal cells, ALA treatment reduced cellular adhesion and invasion [118].

#### 4.4.6. Vitamin C and E

Vitamin C is an essential micronutrient that humans cannot synthesize and must be obtained from the diet. It plays a crucial role in enhancing cellular functions within both the innate and adaptive immune systems and acts as a powerful antioxidant [186,187]. Various fruits from around the world, notably kakadu plum from Australia and camu-camu and acerola from South America, have the most vitamin C. Worldwide, star fruit, guava, black currant, kiwi, and strawberries are good sources of Vitamin C. Cruciferous vegetables, particularly broccoli, kale, and peppers, are also high in vitamin C. Interestingly, fermented cabbage (sauerkraut) contains considerably more vitamin C than most fresh vegetables [188].

Vitamin E, the primary lipid-soluble antioxidant in the cellular antioxidant system, must be obtained exclusively through the diet. It plays a crucial role in protecting polyunsaturated fatty acids, cell membranes, and low-density lipoproteins from oxidation caused by free radicals. Common dietary oils, including olive, palm, rice bran, and peanut, are the most abundant sources of vitamin E [189]. A randomized, placebo-controlled clinical trial was conducted on 60 women aged 15–45 years with pelvic pain and laparoscopic-proven stages 1–3 of endometriosis. Participants received a daily combination of Vitamin C (1000 mg) and Vitamin E (800 IU). After 8 weeks of treatment, there was a significant reduction in pain on the visual analog scale (VAS) scale used to quantify pain compared to the placebo group. Additionally, the severity of dysmenorrhea and dyspareunia significantly decreased in the treatment group [119]. This could potentially be due to the antioxidant status of vitamins C and E and their potential in reducing ROS in endometriosis [119].

#### 4.4.7. Fish Oil (Omega 3 Polyunsaturated Fatty Acids (PUFA))

Fatty acids are fat-soluble components found in plants and animals that serve as the primary building blocks of lipids. They can be either saturated or unsaturated. Unsaturated fatty acids are further classified into monounsaturated and polyunsaturated fatty acids (PUFAs). There are two types of PUFAs, known as omega-3 (ω-3) and omega-6 (ω-6), which are named based on the position of the last double bond from the end of the molecule. The human body can make most fatty acids, except for two essential ones: linoleic acid (LA), an omega-6 fatty acid, and alpha-linolenic acid (ALA), an omega-3 fatty acid. These essential fatty acids must be obtained through the diet because the body cannot produce them. LA and ALA are the simplest forms of omega-6 and omega-3 PUFAs, respectively. Some sources of omega-3 include chia, flax seeds, salmon, tuna, and other seafood, like algae and krill [190,191].

Omega-3 PUFA exhibits inhibitory effects on the conversion of arachidonic acid (AA) into pro-inflammatory compounds, eicosanoids prostaglandin E_2_ (PGE_2_) and leukotriene B_4_ (LTB_4_), which are associated with pelvic pain in endometriosis [192,193]. Women (12–25 years old) who had pelvic pain and endometriosis confirmed by surgery participated in a six-month double-blind, randomized, placebo-controlled study. Of them, 20 received fish oil, 27 received Vitamin D, and 22 received a placebo. Although there was an improvement in VAS pain among those exposed to fish oil, it was only slightly greater than in the other two groups and did not show a statistically significant change throughout the six-month follow-up [112].

Two groups of 120 students with mild to severe dysmenorrhea were randomly assigned to either receive 1000 mg of fish oil capsules daily throughout their menstrual cycle or to take ibuprofen at the onset of pain. The findings suggest that fish oil is more effective than ibuprofen in treating severe pain associated with primary dysmenorrhea [194]. However, it is important to note that dysmenorrhea in women without endometriosis may vary in pain intensity compared to women with endometriosis. Further trials specifically investigating the effects of fish oil in women with endometriosis are needed.

In addition, increased exposure to eicosapentaenoic acid (an ω-3Polyunsaturated Fatty Acid, PUFA) significantly reduces the in vitro survival of endometrial cells compared to cells cultured in media lacking PUFAs or with low or normal ω-3: ω-6 PUFA ratios [121]. This effect may be attributed to the reduction of the inflammatory response, modulation of cytokine function, and decreased prostaglandin production caused by omega-3 [121].

#### 4.4.8. Folate

Folate, or vitamin B9, plays a crucial role in DNA synthesis and repair; however, excessive consumption may have deleterious effects [195]. Women with endometriosis (“benign lesions”) have a higher risk of developing ovarian cancer [196,197]. A recent study has found that women with endometriosis who had higher dietary intake of folate, particularly synthetic folate, had an increased risk of developing invasive ovarian cancer [198]. The association between the consumption of synthetic folate and the development of ovarian cancer was not observed in women without endometriosis [198]. This suggests that synthetic folate might play a role in cancer risk for women with endometriosis.

A potential explanation may be due to the increased susceptibility to genetic mutations, such as MTHFR, in infertile women with endometriosis [51]. The presence of MTHFR polymorphisms, like C677T, can impair the body’s ability to metabolize folic acid effectively, leading to elevated oxidative stress [51]. Oxidative stress plays a role in cellular damage, contributing to inflammation, a key feature in the development of endometriosis, endometriosis-related infertility, and cancer [196,199,200].

#### 4.4.9. Resveratrol

Resveratrol, one of the most researched polyphenols, is found in a variety of plants, including grapes, strawberries, pistachios, mulberries, peanuts, rhubarb, and others. Resveratrol has been suggested as a treatment for endometriosis based on its anti-proliferative, anti-inflammatory, anti-neoplastic, and antioxidant characteristics [122,192]. Resveratrol reduces oxidative stress and cytokine dysregulation and may play a role in reducing the growth of endometriotic tissues, which are all key factors integral to the pathogenesis of endometriosis [192]. Resveratrol reduced MMP-2 and MMP-9 protein and mRNA expression in an in vivo investigation [201]. Moreover, resveratrol supplementation showed positive effects in animal models of endometriosis, reducing the size and quantity of endometrial implants as well as inhibiting growth, vascularization, and inflammation. Nonetheless, it should be mentioned that the resveratrol dosages tested in animal experiments were rather high at 10 mg/kg and 100 mg/kg [122]. The hepatocyte growth factor promotes cell invasion, metastasis, and proliferation, a pivotal finding in endometriosis progression [202]. It has been shown that resveratrol reduced this expression as well as IGF-1 [203].

Additionally, a clinical trial including 12 women who failed to achieve pain alleviation while using an oral contraceptive containing drospirenone and ethinylestradiol examined the impact of resveratrol on the management of endometriosis-related pain. With the addition of 30 mg of resveratrol to the contraceptive regimen, pain scores were significantly reduced, and after two months of treatment, 82% of patients reported that their dysmenorrhea and pelvic discomfort had completely disappeared [122,123]. However, the main limitations of this study were the number of participants, which was limited to only 12 women. Therefore, while resveratrol looks promising, larger randomized control studies are needed.

#### 4.4.10. Quercetin

Quercetin, an important antioxidant derived from a plant pigment, belongs to a class called flavanols. One of the most prevalent dietary flavonoids, it is present in a wide variety of foods, including olive oil, many seeds, buckwheat, nuts, flowers, bark, broccoli, apples, onions, green tea, red grapes, dark cherries, and berries like blueberries and cranberries. It is primarily found in citrus fruits [204]. The highest concentrations of flavanols were found in apples, cherries, berries, onions, and broccoli [205]. Quercetin has been shown to reduce inflammatory mediators, such as prostaglandins and leukotrienes [205]. Furthermore, in a study conducted by Park et al. on mice, quercetin dramatically reduced endometrial cell proliferation and had apoptotic effects on endometrial lesions [206]. Another study found that the combination of quercetin and metformin may enhance autophagy in ectopic endometrial tissues. Moreover, through their combined anti-inflammatory and anti-estrogenic properties, endometrial implants were regressed [124]. In a separate study, quercetin, in a dose-dependent manner and within the ectopic endometrial stromal cells, could inhibit the proliferation, migration, and invasion of endometrial cells. While this study was conducted in vitro from patients with adenomyosis who underwent a hysterectomy, it is well understood that quercetin has anti-proliferative properties and thus could have similar effects in endometriosis [207]. Fadin et al. [208] evaluated the efficacy of a combination supplement including quercetin, *N*-acetyl-cysteine (NAC), and turmeric in 33 women over two months. The results showed a significant improvement in pain [208]. Furthermore, histamine release, FcεRI expression, and IL-6 production are all inhibited by quercetin [209]. Additionally, quercetin reduces the production of all pro-inflammatory substances in granules, including tumor necrosis factor (TNF), by inhibiting MC degranulation [210].

## 5. Conclusions

Several diets and nutrients have been evaluated in patients with endometriosis, with promising outcomes in reducing pain perception as well as overall symptoms of endometriosis. From our review of the literature, it remains inconclusive if a single dietary and nutritional intervention is most appropriate as an adjuvant therapy for endometriosis. A personalized approach to selecting an appropriate dietary and nutritional intervention may be useful, one that considers a comprehensive clinical and lifestyle history of a patient with endometriosis. There is need for further research, by way of well-designed randomized controlled trials, to support evidence-based dietary recommendations for the management of endometriosis and the development of clinical guidelines that can be adopted by clinicians. In addition, large-scale studies on diets, such as the Mediterranean and FODMAP diets, would be useful.

Major gaps still exist in our understanding of how specific diets and nutrients influence the development of endometriosis on a molecular level. We need to understand how food influences estrogen levels, microbiome, and inflammation to promote the perfect storm of pathophysiological mechanisms that initiate and/or progress endometriosis. Nonetheless, there is utility in personalized nutritional counseling and adopting a multi-disciplinary and holistic approach to the management and care of patients with endometriosis, one that includes dieticians, nutritionists, and naturopaths [131,211].

## Figures and Tables

**Figure 1 nutrients-16-03988-f001:**
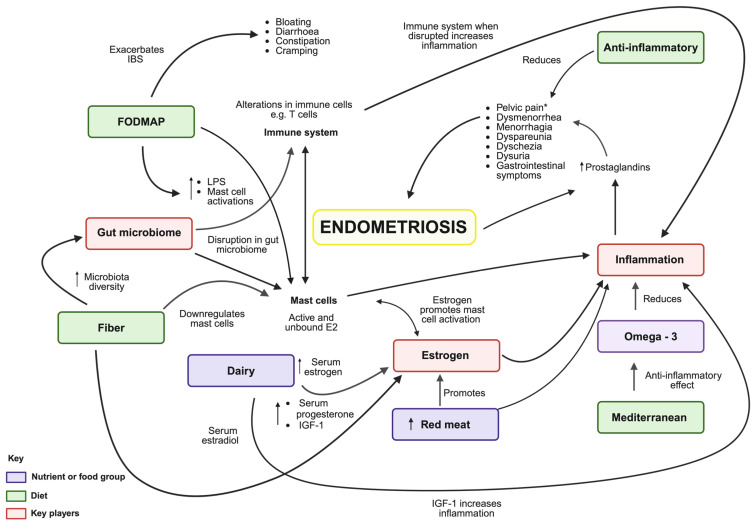
The role diet and nutrients play in the pathophysiology of endometriosis. Diets, food groups, and single nutrients may influence pathophysiological processes underlying endometriosis, such as inflammation, estrogen pathways, and microbiome interactions. A high-fiber diet increases microbiome diversity, which leads to the downregulation of MCs and inflammation. A diverse microbiome metabolizes circulating estrogen. A high FODMAP diet increases IBS-like symptoms [89], such as bloating, which can further worsen the symptoms of endometriosis. High FODMAP increases LPS and MC activation, increasing prostaglandins and leading to pain. Red meat increases inflammation as well as sex hormones and estrogen in the blood, influencing prostaglandins, increasing inflammation, and therefore, pain [90]. The Mediterranean diet, which is rich in antioxidants and ω-3 FAs, reduces inflammation, which has the potential to reduce prostaglandins, and endometriosis-related pain [91]. The anti-inflammatory diet has the same influence. Dairy can increase serum estrogen, progesterone [92], and IGF-1 [93,94,95], all of which influence the levels of estrogen in the blood, which can activate MCs, therefore worsening pain-related symptoms. Estrogen may cause and/or promote MC activation, leading to inflammation. * Increased prostaglandins due to inflammation may account for the pelvic pain typically associated with endometriosis. It is worth noting that patients with endometriosis may present with a wide range of symptoms to varying degrees over the course of their menstrual cycle. These include dysmenorrhea, menorrhagia, dyspareunia, dyschezia, dysuria, infertility, fatigue, anxiety, and depression [96].

**Table 1 nutrients-16-03988-t001:** Dietary interventions evaluated in endometriosis.

Intervention	Effects	Mechanisms	References
Low-nickel diet	↓ Chronic pelvic pain, dysmenorrhea, and dyspareunia	↓ Inflammation	Borghini et al., 2020 [101]
FODMAP diet	↓ Gut symptoms ↓ IBS symptoms associated with endometriosis	Gut barrier restoration Mast cell stabilization	Moore et al., 2017 [18]
Gluten-free diet	↓ Pelvic pain ↓ Abdominal pain	↓ Inflammation	Marziali et al., 2012 [102] Losurdo et al., 2017 [103]
Mediterranean diet	↓ Dyspaeurnia ↓ Dyschezia ↓ Dysmenorrhea	↑ Antioxidant activity ↑ Anti-inflammatory effects	Nirgianakis et al., 2022 [36]
High-fiber diet	↓ Risk of endometriosis ↓ Serum estrogen concentration ↑ Mast cell activation ↑ Microbiome diversity	↓ Estrogen ↑ Microbiome modulation	Parazzini et al., 2013 [104] Aubertin et al., 2008 [105] Rose et al., 1991 [106] Kudesia et al., 2021 [107] Craig et al., 2021 [108] Aleksandrova et al., 2021 [109] Baker et al., 2017 [86]
Anti-inflammatory diet	↓ Preterm birth ↓ Improved health outcome ↓ Reduced risk of pre-eclampsia	↓ Inflammatory markers	Kyozuka et al., 2021 [110]

↑—increase, ↓—decrease.

**Table 2 nutrients-16-03988-t002:** Nutritional interventions evaluated in endometriosis.

Intervention	Effects	Mechanisms	References
Vitamin D	↓ Pelvic pain ↓ C-reactive protein ↑ Total antioxidant capacity Immunomodulatory Inhibit angiogenesis Anti-inflammatory	↓ Pain ↓ Vitamin D modulation	Abolfazel et al., 2021 [111] Nodler et al., 2020 [112] Kalaitzopoulos et al., 2020 [113]
Curcumin	↓ Endometrial cell proliferation by ↓E2 production. TNF-, ↓IL-6, ↓IL-8	↓ E2 ↑ Anti-inflammatory effect	Zhang et al., 2013 [114] Kim et al., 2012 [115]
*N*-acetyl-cysteine (NAC)	↓ Pelvic pain, dysmenorrhea, and dyspareunia Ovarian endometrioma/cyst	↓ Cyst ↓ Pain	Porpora et al., 2013 [116]
Alpha lipoic acid (ALA)	↓ Oxidative stress, endometrial implants. ↓ TNF-α levels in serum and peritoneal fluid ↓ Cellular adhesion and invasion	↓ Inflammatory marker	Pinar et al., 2017 [117] Nicuolo et al., 2021 [118]
Vitamin C and E (combined therapy)	↓ Dysmenorrhea ↓ Dyspareunia ↓ Pelvic pain	↑ Antioxidant activity	Amini et al., 2021 [119]
Fish Oil (Omega 3PUFA)	↓ Pain ↓ Size of lesions Anti-inflammatory	↑ Anti-inflammatory effect ↓ Prostaglandin	Nodler et al., 2020 [112] Tomio et al., 2013 [91] Herington et al., 2013 [120] Gazvani et al., 2001 [121]
Resveratrol	↑ Anti-inflammatory ↑ Antioxidant ↑ Anti-proliferative	↓ Matrix metalloproteinases ↓ Pain	Novakovic et al. [122]
Resveratrol and Drospiernone/ Ethinylestradiol	↓ Pain ↓ Dysmenorrhea	↓ Pain	Maia et al., 2022 [123]
Quercetin	↓ Prostaglandins ↓ Leukotrienes ↓ Endometrial cell proliferation ↓ Anti-inflammatory Anti-proliferative	↓ Proliferation ↓ Prostaglandin ↓ Leukotriene	Jamali et al., 2021 [124]

↑—increase, ↓—decrease.

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
