# Peer review of "Dietary and Nutritional Interventions for the Management of Endometriosis"

_nutrients, 2024, doi:10.3390/nu16233988_

Round 1

Reviewer 1 Report

Comments and Suggestions for Authors

Overview

The manuscript is well-written and -organized.

Critical comments

1   Alternative Healthy Eating Index may be worth incorporating into the manuscript, as it was shown to be associated with a 13% reduction in endometriosis diagnosis in a prospective study/

(Ref. Marcelle M Dougan et al., Am J Obstet Gynecol. 2024 Apr 29:S0002-9378(24)00556-8. A prospective study of dietary patterns and the incidence of endometriosis diagnosis)

2  High dietary folate intake may also be considered in the manuscript as it was shown to be associated with an increased risk of ovarian cancer in women with endometriosis [OR, 1.37 (1.01-1.86)] but not in those without endometriosis in a case-control study

(Ref. Kate Gersekowski et al., Cancer Epidemiol Biomarkers Prev 2023 Aug 1;32(8):1087-1096.)

3  More discussion about whole grain may be added to the manuscript.

(Ref67 Parazzini, F., et al. Selected food intake and risk of endometriosis. Hum Reprod 2004;19(8), 1755-1759 etc).

Author Response

Title: Dietary and nutritional interventions for the management of endometriosis

We thank the reviewers for considering our manuscript and for their valuable comments. We have responded in blue, below.

Response to Reviewer 1 Comments

Overview

The manuscript is well-written and -organized. Thank you for the positive feedback.

Critical comments

1   Alternative Healthy Eating Index may be worth incorporating into the manuscript, as it was shown to be associated with a 13% reduction in endometriosis diagnosis in a prospective study/

(Ref. Marcelle M Dougan et al., Am J Obstet Gynecol. 2024 Apr 29:S0002-9378(24)00556-8. A prospective study of dietary patterns and the incidence of endometriosis diagnosis)

Thank you for this feedback. This study was published after our initial draft, and we welcome its inclusion into this review.

We have referred to it on line 99:

“A recent study, showed that adherence to a healthier dietary pattern, higher in fruits and vegetables and lower in red meat and trans fats (Alternative Healthy Eating Index), was associated with a 13% lower risk of endometriosis diagnosis, and most likely impacts pelvic pain”.

2  High dietary folate intake may also be considered in the manuscript as it was shown to be associated with an increased risk of ovarian cancer in women with endometriosis [OR, 1.37 (1.01-1.86)] but not in those without endometriosis in a case-control study

(Ref. Kate Gersekowski et al., Cancer Epidemiol Biomarkers Prev 2023 Aug 1;32(8):1087-1096.)

Thank you for this suggestion. We have now included the following paragraphs on folate Line 535:

“Folate, or vitamin B9, plays a crucial role in DNA synthesis and repair, however excessive consumption may have deleterious effects128. Women with endometriosis (“benign lesions”) have a higher risk of developing ovarian cancer129,130. A recent study has found that women with endometriosis who consumed higher dietary intake of folate, particularly synthetic folate, had an increased risk of developing invasive ovarian cancer131. The association between consumption of synthetic folate and development of ovarian cancer was not observed in women without endometriosis131. This suggests that synthetic folate might play a role in cancer risk for women with endometriosis.

A potential explanation may be due to the increased susceptibility to genetic mutations such as MTHFR in infertile women with endometriosis132. The presence of MTHFR polymorphisms, like C677T, can impair the body's ability to metabolize folic acid effectively, leading to elevated oxidative stress132. Oxidative stress plays a role in cellular damage, contributing to inflammation, a key feature in the development of endometriosis, endometriosis-related infertility and cancer129,133,134.”

3  More discussion about whole grain may be added to the manuscript.

(Ref67 Parazzini, F., et al. Selected food intake and risk of endometriosis. Hum Reprod 2004;19(8), 1755-1759 etc).

Thank you for this suggestion. Whole grains form significant part of the Mediterranean Diet, the Anti-inflammatory Diet and the High Fibre Diet which have been covered in the review.

We have incorporated the suggested reference from 2004 as per the reviewers suggestion on line 247 “Interestingly, an earlier study of Italian patients with laparoscopically confirmed endometriosis found that consumption of milk,  liver,  carrots,  cheese,  fish  and  whole-grain  foods,  as  well as  coffee  and  alcohol  consumption, were not significantly related to endometriosis48.

Reviewer 2 Report

Comments and Suggestions for Authors

The present paper discussed first the three key players in the pathogenesis of endometriosis: inflammation, estrogen, and the microbiome and then summarized how diet and nutrition could influence their mechanisms, and consequently, the progression and manifestation of endometriosis.

Overall, the paper is well-written, the information covers most of the relevant literature.

However, the following are my concerns about the manuscript that need to be addressed before being considered for publication in Nutrients journal.

First of all, authors didn’t mention anything about DNA methylation and epigenetic mechanism, since aberrant DNA methylation has been reported in endometriotic tissue in genes implicated in the hormonal and inflammatory factors of endometriosis pathogenesis. Aberrant DNA methylation can be amplified by oxidative stress caused by hyperhomocysteinemia that in most cases can be due to incorrect eating habits or to disorders of folate metabolism caused by methylenetetrahydrofolate reductase (MTHFR) gene polymorphisms.

Second, in the paragraph 4.4.7. Fish Oil (Omega 3 Polyunsaturated Fatty Acids (PUFA), you mentioned omega 3 and omega 6 fatty acids. These two essential fatty acids have different properties: the first one is the precursor of EPA and DHA, while the other is the precursor of arachidonic acid, different eicosanoids, different roles. Please, rewrite the paragraph taking into consideration this knowledge.

Minor:

Some examples, but more all over the text

Line 19: the pathogenesis of endometriosis; (use colons instead of semicolon)

Line 19: correct summarise into summarize

Line 485 and 487: w-3 and not Ω-3

Comments on the Quality of English Language

English is fine, some typos, grammatical errors (all over the manuscript, here are some examples)

Line 19: the pathogenesis of endometriosis; (use colons instead of semicolon)

Line 19: correct summarise into summarize

Line 485 and 487: w-3 and not Ω-3

Author Response

The present paper discussed first the three key players in the pathogenesis of endometriosis: inflammation, estrogen, and the microbiome and then summarized how diet and nutrition could influence their mechanisms, and consequently, the progression and manifestation of endometriosis.

Overall, the paper is well-written, the information covers most of the relevant literature.

However, the following are my concerns about the manuscript that need to be addressed before being considered for publication in Nutrients journal.

First of all, authors didn’t mention anything about DNA methylation and epigenetic mechanism, since aberrant DNA methylation has been reported in endometriotic tissue in genes implicated in the hormonal and inflammatory factors of endometriosis pathogenesis. Aberrant DNA methylation can be amplified by oxidative stress caused by hyperhomocysteinemia that in most cases can be due to incorrect eating habits or to disorders of folate metabolism caused by methylenetetrahydrofolate reductase (MTHFR) gene polymorphisms.

Thank you for this suggestion. We agree with the reviewer and have made the following additions to the manuscript:

Line 157 “Oxidative stress can amplify DNA methylation alterations, implicated in the pathogenesis of endometriosis29. Oxidative stress caused by hyper-homocysteinemia can be triggered by diets high in methionine (precursor of homocysteine)-rich foods like red meat and dairy or disorders of folate metabolism caused by methylenetetrahydrofolate reductase (MTHFR) gene polymorphisms. MTHFR C677T homozygous polymorphisms might be considered a risk factor for endometriosis3031.”

Second, in the paragraph 4.4.7. Fish Oil (Omega 3 Polyunsaturated Fatty Acids (PUFA), you mentioned omega 3 and omega 6 fatty acids. These two essential fatty acids have different properties: the first one is the precursor of EPA and DHA, while the other is the precursor of arachidonic acid, different eicosanoids, different roles. Please, rewrite the paragraph taking into consideration this knowledge.

Thank you for highlighting this. We have revised this paragraph and ensured clarity around our discussion of the omega 3 fish oil intervention for endometriosis.

Line 526 “Fatty acids are fat-soluble components found in plants and animals and serve as the primary building blocks of lipids. They can be either saturated or unsaturated. Unsaturated fatty acids are further classified into monounsaturated and polyunsaturated fatty acids (PUFAs). There are two types of PUFAs, known as omega-3 (ω-3) and omega-6 (ω -6), which are named based on the position of the last double bond from the end of the molecule. The human body can make most fatty acids, except for two essential ones: linoleic acid (LA), an omega-6 fatty acid, and alpha-linolenic acid (ALA), an omega-3 fatty acid. These essential fatty acids must be obtained through the diet because the body cannot produce them. LA and ALA are the simplest forms of omega-6 and omega-3 PUFAs, respectively. Some sources of omega-3 include chia, flax seeds, salmon, tuna, and other seafood like algae and krill 129.

Omega-3 PUFA exhibits inhibitory effects on the conversion of arachidonic acid (AA) into pro-inflammatory compounds eicosanoids prostaglandin E2 (PGE2) and leukotriene B4 (LTB4), which are associated with pelvic pain in endometriosis130131.”

Minor:

Some examples, but more all over the text

Line 19: the pathogenesis of endometriosis; (use colons instead of semicolon)

This has now been corrected.

Line 19: correct summarise into summarize

This has now been corrected.

Line 485 and 487: w-3 and not Ω-3

This has now been corrected to ω-3.

Comments on the Quality of English Language

English is fine, some typos, grammatical errors (all over the manuscript, here are some examples)

Line 19: the pathogenesis of endometriosis; (use colons instead of semicolon)

Line 19: correct summarise into summarize

Line 485 and 487: w-3 and not Ω-3

Thank you, these have been corrected.

Reviewer 3 Report

Comments and Suggestions for Authors

In this manuscript entitled "Dietary and nutritional interventions for the management of endometriosis", The authors discussed three key players in the pathogenesis of endometriosis; inflammation, estrogen, and the microbiome and summarised how diet and nutrition can influence their mechanisms, and consequently, the progression and manifestation of endometriosis. This review analysed and summarised the mechanisms of diet and nutrition in terms of their influence on endometriosis, which is innovative and enlightening for the adjuvant treatment of this disease. In conclusion, I think it can be accepted after solving the following problems.

1. The review lacks diagrams or tables, and it is suggested that some diagrams could be cited or created to enhance the readability and visualisation of the article. For example, a diagram of the pathogenesis of endometriosis or a table of dietary and nutritional interventions should be added.

2. Pay attention to the correct use of tenses, e.g., in lines 18 and 86, discuss should be changed to discussed.

3. Lines 219-220 referred to tables 1 and 2, but the article did not present these tables, please explain and complete.

4. Figure 1 on line 230 was missing, please add it.

5. With regard to dietary interventions, the authors mentioned the Mediterranean diet, the high-fibre diet and the anti-inflammatory diet, etc., were these concepts taken from authoritative sources? Because these concepts were not clearly defined, and at the same time they were related to each other.

6. What might be the mechanism by which vitamin D can intervene in endometriosis? Please add in section 4.4.2.

7. By summarising these studies, what insights do the authors gain about dietary and nutritional interventions? Which are the most appropriate for managing endometriosis? If possible, add to the conclusion.

8. The review was written in a logical manner, but it would have been better to dig deeper into the correlation between the intervening factors and the treatment of endometriosis from a data perspective. For example, summarise and quantify the effect of the dose of Mediterranean diet and the type and dose of antioxidants used in the treatment of endometriosis. Because the data would be more conducive to understanding the effect of their interventions.

Comments on the Quality of English Language

Pay attention to the correct use of tenses, e.g., in lines 18 and 86, discuss should be changed to discussed.

Author Response

In this manuscript entitled "Dietary and nutritional interventions for the management of endometriosis", The authors discussed three key players in the pathogenesis of endometriosis; inflammation, estrogen, and the microbiome and summarised how diet and nutrition can influence their mechanisms, and consequently, the progression and manifestation of endometriosis. This review analysed and summarised the mechanisms of diet and nutrition in terms of their influence on endometriosis, which is innovative and enlightening for the adjuvant treatment of this disease. In conclusion, I think it can be accepted after solving the following problems.

  1. The review lacks diagrams or tables, and it is suggested that some diagrams could be cited or created to enhance the readability and visualisation of the article. For example, a diagram of the pathogenesis of endometriosis or a table of dietary and nutritional interventions should be added.

We have included two tables and one figure in the original submission.

Table 1: Dietary interventions evaluated in endometriosis.

Table 2: Nutritional interventions evaluated in endometriosis.

Figure 1: The role diet and nutrients play in the pathophysiology of endometriosis.

  1. Pay attention to the correct use of tenses, e.g., in lines 18 and 86, “discuss” should be changed to “discussed”.

This has now been corrected.

  1. Lines 219-220 referred to tables 1 and 2, but the article did not present these tables, please explain and complete.

We have included two tables in the original submission.

  1. Figure 1 on line 230 was missing, please add it.

Added.

  1. With regard to dietary interventions, the authors mentioned the Mediterranean diet, the high-fibre diet and the anti-inflammatory diet, etc., were these concepts taken from authoritative sources? Because these concepts were not clearly defined, and at the same time they were related to each other.

We agree with the reviewer these diets are related, and do overlap but they are distinct and have been examined as separate interventions for the management of endometriosis, as detailed under each section within the manuscript.

  1. What might be the mechanism by which vitamin D can intervene in endometriosis? Please add in section 4.4.2.

We have added this on Line 446: In endometriosis, vitamin D may decrease inflammation, immunoregulation, and inhibit angiogenesis114 (see Figure 2).

  1. By summarising these studies, what insights do the authors gain about dietary and nutritional interventions? Which are the most appropriate for managing endometriosis? If possible, add to the conclusion.

Thank you for this suggestion. We have amended our conclusion and added the following:

Line 625: “From our review of the literature, it remains inconclusive if a single dietary and nutritional intervention is most appropriate as adjuvant therapy for endometriosis. A personalised approach to selecting an appropriate dietary and nutritional intervention is required, one that considers a full clinical and lifestyle history of a patient with endometriosis”

  1. The review was written in a logical manner, but it would have been better to dig deeper into the correlation between the intervening factors and the treatment of endometriosis from a data perspective. For example, summarise and quantify the effect of the dose of Mediterranean diet and the type and dose of antioxidants used in the treatment of endometriosis. Because the data would be more conducive to understanding the effect of their interventions.

Thank you for this suggestion. We agree with the reviewer and are interested in this data too. We are currently drafting a deeper systematic review that will dig deeper to this level. The purpose of this scoping review was to be a precursor for a systematic review, to highlight the main gaps and give a high-level overview of the main dietary and nutritional interventions for adjuvant therapy of endometriosis.

Comments on the Quality of English Language

Pay attention to the correct use of tenses, e.g., in lines 18 and 86, “discuss” should be changed to “discussed”.

This has now been corrected throughout.

Round 2

Reviewer 1 Report

Comments and Suggestions for Authors

The manuscript improved very much.

Author Response

Thank you for your time and for your revisions to improve the paper.